# Single-Cell RNA-Seq Analysis Reveals Ferroptosis in the Tumor Microenvironment of Clear Cell Renal Cell Carcinoma

**DOI:** 10.3390/ijms24109092

**Published:** 2023-05-22

**Authors:** Jing Zhang, Yun Deng, Hui Zhang, Zhiyuan Zhang, Xin Jin, Yan Xuan, Zhen Zhang, Xuejun Ma

**Affiliations:** 1Department of Radiation Oncology, Fudan University Shanghai Cancer Center, Shanghai 200032, China; 10301010083@fudan.edu.cn (J.Z.); 15111230011@fudan.edu.cn (Y.D.); huizhang12@fudan.edu.cn (H.Z.); 12307120021@fudan.edu.cn (Z.Z.); 15301050228@fudan.edu.cn (X.J.); yxuan16@fudan.edu.cn (Y.X.); 2Department of Oncology, Shanghai Medical College, Fudan University, Shanghai 200032, China; 3Shanghai Clinical Research Center for Radiation Oncology, Shanghai 200032, China; 4Shanghai Key Laboratory of Radiation Oncology, Shanghai 200032, China

**Keywords:** ferroptosis, microenvironment, renal cell carcinoma, single-cell, prognosis

## Abstract

In this study, we investigated the role of ferroptosis in the tumor microenvironment (TME) of clear cell renal cell carcinoma (ccRCC), the leading cause of renal cancer-related death. We analyzed single-cell data from seven ccRCC cases to determine cell types most correlated with ferroptosis and performed pseudotime analysis on three myeloid subtypes. We identified 16 immune-related ferroptosis genes (IRFGs) by analyzing differentially expressed genes between cell subgroups and between high and low immune infiltration groups in the TCGA-KIRC dataset and the FerrDb V2 database. Using univariate and multivariate Cox regression, we identified two independent prognostic genes, AMN and PDK4, and constructed an IRFG score model immune-related ferroptosis genes risk score (IRFGRs) to evaluate its prognostic value in ccRCC. The IRFGRs demonstrated excellent and stable performance for predicting ccRCC patient survival in both the TCGA training set and the ArrayExpress validation set, with an AUC range of 0.690–0.754, outperforming other commonly used clinicopathological indicators. Our findings enhance the understanding of TME infiltration with ferroptosis and identify immune-mediated ferroptosis genes associated with prognosis in ccRCC.

## 1. Introduction

An estimated 3% of adult malignant tumor deaths are caused by renal cell carcinoma (RCC), one of the three most common genitourinary malignancies [1]. Approximately 75% of RCC patients have clear cell renal cell carcinoma (ccRCC) [2,3]. Approximately 20–30% of patients with ccRCC are incidentally diagnosed with metastatic RCC, and approximately 30% of patients with early-stage ccRCC will develop distant metastases within five years after surgery [4]. Patients with metastatic ccRCC have a poor prognosis, with a five-year survival rate of 10% [5]. A variety of therapeutic modalities are used to treat ccRCC, including targeted therapy and immunotherapy [6]. Metastasis-directed therapy, such as radiation and surgery, is also a potential therapeutic modality for ccRCC [7]. However, these clinical strategies have limitations due to patient heterogeneity, which can result in frequent side effects and drug resistance. Tumor heterogeneity may also contribute to these limitations, especially in terms of drug resistance [5]. Therefore, it is imperative to explore the molecular mechanisms underlying the initiation and progression of ccRCC, identify novel targets, and develop accurate prognostic systems.

CcRCC is associated with ferroptosis, a form of programmed cell death that is iron- and lipid-dependent [8]. It is important to note that despite advances in ccRCC therapy, some patients remain resistant to treatment due to mechanisms that inhibit apoptosis [9,10]. Recently, several molecules have been reported to be involved in ferroptosis during ccRCC progression. Wang et al. [11] found that ccRCC tumors frequently express SUV39H1, which induces iron accumulation and lipid peroxidation, resulting in ferroptosis. Another study showed that ccRCC cells were inhibited by KLF2 during migration and invasion by regulating ferroptosis through GPX4 [12]. Li et al. [13] reported that inhibition of SLC7A11 leads to ferroptosis in kidney cancer. Ferroptosis has been shown to be beneficial in the treatment of ccRCC in several studies [11,14]. Importantly, these studies emphasize the role of ferroptosis in suppressing tumor growth and enhancing therapeutic efficacy. Yang et al. demonstrated that the Hippo pathway effector TAZ regulates ferroptosis in ccRCC, suggesting a potential therapeutic target [14]. Similarly, Wang et al. showed that SUV39H1 deficiency suppresses ccRCC growth by inducing ferroptosis [11]. These findings collectively suggest that targeting the process of ferroptosis holds promise as an effective therapeutic strategy for ccRCC. However, it remains unclear how ferroptosis-associated genes are expressed in the tumor microenvironment (TME), including immune cells that infiltrate the tumor.

Since tumors reside in a heterogeneous microenvironment composed of a variety of cell types, immune cells infiltrating tumors play an important role in influencing prognosis and response to immunotherapies [15]. For the identification of tumor cellular characteristics, bulk sequencing is not appropriate. Single-cell sequencing could identify cell populations whose gene expression patterns may have been masked or diluted by bulk sequencing [16,17]. By characterizing the cellular composition and transcriptional state of ccRCC, single-cell RNA sequencing (scRNA-seq) has been used to investigate the origin and intratumoral heterogeneity of this cancer [18,19]. Furthermore, scRNA-seq analysis revealed a variety of cell populations associated with immunotherapy resistance and poor prognosis in patients with ccRCC [20,21,22]. It is still unclear how ferroptosis regulates these changes within the TME of ccRCC, despite significant progress in identifying transcriptional alterations associated with cancer development and clinical treatment.

To investigate the association between ferroptosis-related genes and tumor-infiltrating immune cells in the microenvironment, we integrated scRNA-seq (GSE159115) and The Cancer Genome Atlas (TCGA) (KIRC) data. We identified specific immune-related ferroptosis genes and then developed prognostic models based on these genes. According to our multiomics analysis focused on the cellular and molecular levels, ferroptosis-related score is associated with immune-cell infiltration and immune-related pathways. These findings not only contribute to our understanding of ferroptosis in the tumor microenvironment (TME) but also have the potential to enhance existing prognostic biomarkers in this context, thereby improving patient prognosis. Furthermore, these findings may also provide new therapeutic possibilities for the treatment of ccRCC.

## 2. Results

### 2.1. Identification of ccRCC Single-Cell Subpopulations

The comprehensive analysis workflow is shown in Figure 1. According to the quality control criteria, we performed quality control, dimensionality reduction, and clustering on seven clear cell renal cell carcinoma samples from GSE159115. The clinical information is summarized in Table 1. The quality control results were satisfactory and showed a high correlation between nCount and nFeature (Figure 2D). We used the harmony method to eliminate batch effects, and as observed in Figure 2C, the removal of batch effects was substantial, with sample distributions present in each cell cluster. Using a resolution of 1.6 as the clustering criterion, we then classified the 31 subpopulations into nine distinct cell types (Figure 2A,B) based on signature markers derived from single-cell correlation papers (Figure 2E; Appendix A): epithelium, endotheliocytes, fibroblasts, myeloid, T cells, proliferating cells, NKT cells, B cells, and mast cells.

### 2.2. Ferroptosis Activity in Different Immune Cell Types

We performed differential analysis on all cell types and presented the intersection between differentially expressed genes (adj. *p* value < 0.05, |log2FC| > 0.5, Appendix A) across cell types and ferroptosis-related genes (Figure 3A). We found that the differentially expressed ferroptosis-related genes in T cells, NKT cells, and B cells were mostly downregulated, such as HMOX1, PEBP1, PDK4, EGR1, TIMP1, IL1B, and TIMP1. In contrast, myeloid showed the highest number of differentially expressed ferroptosis-related genes, with a significant number of both upregulated and downregulated genes. Upregulated genes included IL1B, FTL, CTSB, SLC40A, and CYBB, while downregulated genes included COX4I2, WWTR1, CP, EGR1, and ENPP2. To determine which immune cell type had the highest association with ferroptosis, we used the AUCell package to calculate the ferroptosis pathway activity in each cell. As shown in Figure 3B, ferroptosis pathway activity varied between different immune cells. Myeloid showed the highest ferroptosis activity, while T cells showed the lowest activity.

### 2.3. Subtyping of Myeloid

Since the ferroptosis pathway showed the highest activity in myeloid, we extracted myeloid subgroups for dimensionality reduction and clustering. We then used the harmony method to remove batch effects, and as shown in Figure 3E, the batch effect removal was evident, with a relatively uniform distribution of samples across cell clusters. We then set the resolution to 0.2 as the clustering criterion and classified the seven subgroups into three cell types—monocytes, macrophages, and dendritic cells—based on the collection of myeloid marker genes from the single-cell literature (Appendix A and Figure 3C,D).

To explore the enrichment of ferroptosis-related genes in myeloid, we intersected the differentially expressed genes of myeloid relative to other cell groups with ferroptosis-related genes and obtained a total of 62 myeloid-related ferroptosis genes (Figure 4A and Appendix A). GO and KEGG enrichment analysis revealed that the 62 myeloid-related ferroptosis genes were associated with biological processes such as response to hypoxia, iron ion homeostasis, and cellular iron ion homeostasis; with cellular components such as organelle outer membrane, mitochondrial outer membrane, and basal plasma membrane; and with molecular functions such as peroxidase activity, antioxidant activity, and protein heterodimerization activity. KEGG analysis showed that the 62 myeloid-related ferroptosis genes were associated with pathways such as ferroptosis, necroptosis, HIF-1 signaling pathway, and TNF signaling pathway (Figure 4B,C and Appendix A).

To analyze the differentiation of different myeloid subtypes, we used the monocle package to perform pseudotime analysis on the three myeloid subtypes. As shown in Figure 5A–F, with the progression of pseudotime, the number of dendritic cells and monocytes gradually decreased, while macrophages gradually differentiated toward State6 and State7. The decrease in monocytes may be due to their gradual transformation into macrophages. Based on this pseudotime trend, we plotted a heatmap of the expression of the 62 myeloid-related ferroptosis genes as a function of pseudotime (Figure 5G). The heatmap shows that these genes exhibit different expression trends, for example, the expression trends of genes such as CP, CIRBP, and HILPDA increase from low to high; the expression trends of genes such as FTH1, AKR1C3, NDRG1, IL1B, and HIF1A decrease from high to low; while the expression trends of genes such as EGR1, ATF3, AMN, and CA9 first increase and then decrease.

### 2.4. Identification of Immune-Related Ferroptosis Genes (IRFGs)

To identify IRFGs associated with both immune cells and ferroptosis pathways, we used the estimate package to evaluate the immune-cell infiltration score of ccRCC samples in the TCGA dataset, and according to the score, we divided ccRCC samples into high immune-cell infiltration samples and low immune-cell infiltration samples. We then performed differential analysis using the DESeq2 package (Figure 6A,B). With adj. *p* value < 0.05 and |log2FC| > 0.5 as criteria, a total of 5032 differentially expressed genes were identified, including 3272 upregulated genes and 1760 downregulated genes in high immune-cell infiltration samples (Appendix A). To find the IRFGs related to immune cells and ferroptosis pathways, we took the intersection of the differentially expressed genes between high and low immune-cell infiltration samples from bulk RNA-seq data, differentially expressed genes of immune cells compared to other cell types in single-cell data, and ferroptosis-related genes (Figure 6C). In total, we identified 16 IRFGs: IL1B, FTL, CTSB, TNFAIP3, HMOX1, CD44, TMSB4X, CYBB, ALOX5, GDF15, PDK4, AMN, FABP4, NR4A1, IFNG, and GABARAPL1.

### 2.5. Mutation and Methylation Signatures of IRFGs

To further investigate the SNV, CNV, and methylation characteristics of the 16 IRFGs, we downloaded data for 152 SNV samples, 273 CNV samples, and 141 methylation samples from the TCGA dataset. By analyzing the overall CNV alterations in the samples, we found that chromosomal copy number increases in ccRCC mainly occurred in the 5q region, while decreases were mostly observed in the 1p, 2q, and 9q regions (Figure 7B). Analyzing the mutation status of the 16 IRFGs in different samples, we found that most IRFGs did not have gene mutations or chromosomal copy number variations. However, CTSB, FABP4, ALOX5, CD44, and FTL showed gene mutations or chromosomal copy number variations in 121 samples, most of which showed a decrease in chromosomal copy number. In addition, there were no significant differences in chromosomal copy number variations between samples with high and low immune-cell infiltration samples, but gene mutations in CTSB, ALOX5, CD44, and FTL were found in individual samples with low immune-cell infiltration samples, whereas no gene mutations in IRFGs were observed in samples with high immune-cell infiltration samples (Figure 7A). Further comparison of the TMB between high and low immune-cell infiltration samples showed no significant differences (Figure 7C). By analyzing the differences in the 16 IRFGs between high and low immune-cell infiltration samples, we found that the methylation levels of AMN, IFNG, and ALOX5 were increased in high immune-cell infiltration samples, while only CD44 showed a decreased methylation level in high immune-cell infiltration samples (Figure 7D,E).

### 2.6. Impact of IRFGs on ccRCC Prognosis

To facilitate reproducibility of the data, we performed univariate and multivariate Cox regression to investigate the impact of the 16 IRFGs on the overall prognosis of ccRCC patients. Through these analyses, we identified two independent risk factors for ccRCC prognosis, which are the AMN and PDK4 genes. Based on these two genes, we constructed an overall IRFGs score called immune-related ferroptosis genes risk score (IRFGRs). Based on the results in Figure 8, we observed an association between “AMN gene only” and “PDK4 only” with ccRCC prognosis. We first calculated the optimal cut-off value using surv_cutpoint and divided the TCGA training set and ArrayExpress validation set into high- and low-score groups based on the cut-off value, and then plotted the survival curves for each group. The results showed significant differences in prognosis between the two groups (training set *p* value < 0.001, validation set *p* value = 0.031; Figure 9A,B). We then calculated the AUC of IRFGRs in both the training and validation sets and found that the AUC of one-, two-, and three-year training sets were 0.723, 0.690, and 0.730, respectively (Figure 9C), while the AUC of one-, two-, and three-year validation sets were 0.754, 0.743, and 0.696, respectively (Figure 9D). The accuracy of survival prediction in the validation set was similar to that in the training set, and the short-term survival prediction within one–two years was even more accurate than in the training set.

To further investigate the impact of IRFGRs and other common clinicopathological indicators on the overall survival of ccRCC, we performed univariate and multivariate Cox regression analyses on these factors and constructed a forest plot (Figure 10A). The results of the multivariate analysis showed that only IRFGRs, pNstage, pMstage, and age were independent risk factors for the overall survival of ccRCC, with IRFGRs having the most significant impact (Figure 9E). To accurately assess the influence of each independent risk factor and the nomogram on overall survival, we calculated the AUC for each factor. As shown in Figure 10B, the predictive performance and stability of the integrated nomogram were optimal, with AUC values above 0.8 for 1–5 years. The predictive performance and stability of the IRFGRs followed closely, with AUC values above 0.7 for 1–5 years. Clinical decision curve analysis demonstrated similar findings, with the nomogram exhibiting the widest range of net patient benefits and thresholds, followed by IRFGRs. To generated the stability of the nomogram, we conducted concordance curves for 1-, 2-, and 3-year survival (Figure 10D–F), which revealed that the nomogram predictions were in general agreement with the actual outcomes. Taken together, these results underscore the accuracy and stability of the nomogram constructed using IRFGRs for predicting overall survival and patient benefit in ccRCC.

### 2.7. Exploring the Gene Modules and Their Functions Associated with the IRFGRs

Given the significant impact of IRFGRs on the overall survival of ccRCC, we further explored the related gene modules and functions by performing weighted gene co-expression network analysis (WGCNA) on the training set for the gene modules with the highest correlation to the IRFGRs score. No outlier samples were found by cluster analysis (Figure 11A). We then constructed a scale-free network with a soft threshold of 4 and set a minimum of 30 genes per module to form 19 modules (Figure 11B). We then merged similar modules by setting a minimum module distance of 0.2, resulting in a final count of 15 modules (Figure 11C). By calculating the correlation between the modules and clinical features, we created a correlation heatmap (Figure 11D) and selected the pink module with the highest correlation with IRFGRs, as the core module. This module contained 242 genes (Appendix A).

To further explore the gene functions of the pink module, we performed GO and KEGG enrichment analyses. The GO analysis results revealed that the genes in the pink module were related to biological processes such as small molecule catabolic process, xenobiotic export, and organic acid biosynthetic process. They were associated with cellular components such as the basal plasma membrane, apical part of the cell, and basolateral plasma membrane, and with molecular functions such as transaminase activity, organic acid transmembrane transporter activity, and ligand-gated anion channel activity. KEGG analysis showed that the 62 myeloid-related ferroptosis genes were involved in the ECM-receptor interaction, TGF-beta signaling pathway, IL-17 signaling pathway, and focal adhesion (Figure 12A,B and Appendix A).

### 2.8. Investigating the Relationship between IRFGRs and Immune-Cell Infiltration

To further explore the relationship between IRFGRs and immune-cell infiltration, we used CIBERSORT to assess the infiltration scores of 22 immune cell types in ccRCC samples from TCGA. We then used the corrplot package to plot the correlation analysis between the 22 immune-cell infiltration scores and IRFGRs as well as their constituent genes (Figure 12C). From the plot, it can be observed that IRFGRs showed a positive correlation with the infiltration levels of immune cells such as plasma cells, T cells CD4 memory activated, T cells follicular helper, and M2 macrophages, while they showed a negative correlation with the infiltration levels of immune cells such as T cells CD4 memory resting, NK cells resting, and mast cells resting.

## 3. Discussion

Immune checkpoint inhibitors including PD-1/PD-L1 and CTLA4 antibodies have been clinically approved for the treatment of metastatic ccRCC [2,23,24]. Although PD-1 antibodies have been shown to be effective, some patients are still non-reactive to them [25]. Recent studies have shown that the prognosis of ccRCC is influenced by ferroptosis-related genes [26]. There is a crosstalk between ferroptosis and the immune system [27,28,29]. Despite this finding, no study has focused on ferroptosis in the microenvironment, including immune cells, and its prognostic significance, and we hope that our analysis has illustrated this correlation.

Using publicly available single cell RNA-seq data and bulk RNA-seq data, we investigated ferroptosis and its impact on tumor immunity. We found that myeloids had the most differentially expressed FRGs and identified IRFGs associated with immune cells and the ferroptosis pathway. A score derived from these IRFGs could be used to predict the outcome of patients with ccRCC.

In this study, ferroptosis was identified as a susceptibility of ccRCC. There is evidence that ether phospholipids can drive ferroptosis by acting as additional substrates for lipid peroxidation, further indicating their importance in the development of ferroptosis susceptibility in ccRCC [30]. Cheerin has also been shown to suppress fatty acid oxidation, prevent ferroptosis, and maintain fatty acid levels in another multiomics approach [31]. Therefore, therapies targeting ferroptosis may resensitize resistant tumors to immunotherapy in ccRCC patients with specific metabolic features.

In our study, we observed the highest enrichment of “myeloid-related ferroptosis genes” in the context of hypoxia and oxygen level response. According to the study by Green et al., inhibition of ISCA2 reduces the expression of hypoxia-inducible factor (HIF) and induces ferroptosis in ccRCC. This suggests that increased HIF1a activity in ccRCC may be associated with ISCA2 inhibition, beyond just VHL loss [32]. Ferroptosis is a specific cell death pathway characterized by the accumulation of intracellular free iron and increased oxidative stress. Therefore, these pathways can be considered pathways specifically associated with ferroptosis [33].

In the TME of ccRCC, the ferroptosis pathway activity is highest in myeloid, which can be further divided into three types: monocytes, macrophages, and dendritic cells. There was a gradual decrease in dendritic cells and monocytes, and a gradual differentiation of macrophages. Previous studies have reported that tumor-associated macrophages (TAMs) and immunosuppressive M2-polarized macrophages are recruited in patients with increased ferroptosis [34]. One study showed that removing GPX4 from TAMs can reduce the viability of M2 TAMs without affecting M1 TAMs, as ferroptosis resistance was higher in M1 than in M2 [35]. It appears that macrophages are an important component of the immune system and may play a role in ferroptosis. Liu et al. revealed that NCOA4-mediated ferroptosis promotes M2 polarization of macrophages in COPD emphysema. Their study showed that ferroptosis in bronchial epithelial cells, mediated by NCOA4, contributes to the M2 polarization of macrophages [36]. This indicates a potential association between ferroptosis and the polarization state of immune cells. An immunosuppressive TME may be reversed by targeting macrophages with ferroptosis.

Next, we examined the microenvironment for ferroptosis-associated genes, which are significantly associated with poor prognosis in ccRCC patients. In our study, IL1B expression was found to be increased in the myeloids of ccRCC patients. Li et al. [37] investigated the cellular characteristics of renal tumors using single-cell and spatial sequencing and concluded that IL1B signaling from macrophages drives an invasive phenotype at the tumor-normal kidney interface. Furthermore, we found that CTSB and FTL expression was increased in myeloids and that mutations in these genes were associated with low levels of immune infiltration. Chen et al. [38] investigated the role of CTSB in RCC and showed that inhibition of CTSB expression in vitro and in vivo inhibited RCC growth. Li et al. [39] found that the extracellular matrix (ECM) was degraded by CTSB in RCC, which enhanced the ability to invade and metastasize. Hu et al. [40] found that FTL levels were positively associated with tumor infiltration, and Mou et al. [41] found that the interaction of FTL with NCOA4 in ccRCC was correlated with a poor prognosis and impaired immune infiltration. In our study, we found that the methylation levels of ALOX5 and IFNG were positively associated with the level of immune infiltration. Based on the findings from recent studies, the role of the identified genes in the immunotherapy of renal clear cell carcinoma (RCC) appears to be associated with potential benefits rather than detriments. Fang et al. reported that prognosis-related genes, which were found to participate in the immunotherapy of RCC, potentially target dendritic cells. Dendritic cells play a critical role in the initiation of immune responses by presenting antigens to T cells and activating the immune system. Therefore, the involvement of these genes in targeting dendritic cells suggests a positive impact on immunotherapy outcomes [42]. Martini et al. investigated angiogenic and immune-related biomarkers in RCC patients treated with axitinib and pembrolizumab, an immunotherapy combination. The results indicated that certain biomarkers were associated with improved outcomes in terms of response to treatment and overall survival [43]. In addition, CD44, a marker of cancer stem cells, is also associated with iron homeostasis and immune infiltration, as shown in previous studies [44,45,46,47]. Some of the genes, including HMOX1, AMN, and PDK4, were found to be IRFGs. In ccRCC, they have been shown to be independent prognostic predictors of overall survival and correlated with tumor immunity [48,49,50].

However, several limitations remain, and further research is needed. First, there was a lack of racial diversity in the samples, and the number of samples was limited. In addition, a traceability database was used to build and validate the model. The clinical efficacy of ferroptosis in ccRCC has not yet to be proven, and the molecular mechanisms behind its action have not yet been identified. Further experiments and clinical data are needed to determine whether immunoferroptosis genes and renal clear cell carcinoma interact. We plan to address the score in future studies by including a more contemporary cohort that encompasses patients treated with immunotherapy. This will enable us to assess the score’s performance in this specific treatment setting and further validate its clinical utility. Nevertheless, our study provides insight into ferroptosis as a biomarker and therapeutic target in ccRCC using transcriptional and clonotypic analyses of immune cells.

Although our study provides insights into the association of IRFGs with tumor biology, it is crucial to evaluate whether these genes can serve as potential biomarkers for predicting immunotherapy response [51]. It is worth noting that the existing PD-1 has been shown to be a poor biomarker in RCC [52]. Therefore, future evaluations of our model’s predictive value in terms of immunotherapy checkpoint inhibitor response may be necessary, as it remains an unmet need in the field. However, several factors need to be considered to determine its feasibility. This includes the availability of standardized and reproducible testing methods, integration of the model into existing diagnostic frameworks, and the establishment of clinical guidelines for interpreting the results. Additionally, large-scale validation studies involving diverse patient cohorts and different treatment regimens are required to assess the robustness and generalizability of the model. These efforts will ultimately contribute to the translation of our findings into clinical practice.

## 4. Materials and Methods

### 4.1. Single-Cell Data Acquisition

We obtained single-cell sequencing and clinical information datasets of patients with clear cell renal cell carcinoma (ccRCC) from the Gene Expression Omnibus (GEO) database (GSE159115) [53]. This dataset was generated using 10× Genomics technology and sequenced on the Illumina HiSeq 2500 platform. The dataset includes seven ccRCC tissue samples, six normal kidney tissue samples, and one chromophobe renal cell carcinoma sample. In this study, we only included the seven ccRCC tissue samples to focus our single-cell analysis on the tumor tissues.

### 4.2. Acquisition of Ferroptosis-Related Genes

We downloaded 369 ferroptosis-driving genes, 348 ferroptosis-suppressing genes, and 11 ferroptosis marker genes from the FerrDb V2 database (http://www.zhounan.org/ferrdb/current/) (accessed on 25 February 2022) [54]. After merging and removing duplicates, we obtained 484 unique ferroptosis-related genes (FRGs). Detailed information is provided in Appendix A.

### 4.3. Bulk RNA-Seq Data Acquisition

To further integrate single-cell sequencing data with large-scale bulk RNA-seq analysis, we used the TCGAbiolinks package [55] to download ccRCC and adjacent normal tissue samples from TCGA as a bulk RNA-seq training set. The TCGAbiolinks package facilitates the download of the latest 613 ccRCC samples and corresponding clinicopathologic survival information from the Genomic Data Commons (GDC, https://portal.gdc.cancer.gov/) (accessed on 3 April 2022). In addition, we downloaded the E-MTAB-1980 ccRCC dataset [56] from ArrayExpress (https://www.ebi.ac.uk/arrayexpress/) (accessed on 5 April 2022) [57]. This dataset was generated using the Agilent Human Gene Expression 4x44K v2 Microarray platform and contains 104 ccRCC tissue samples and corresponding clinical information.

Our study included bulk RNA-seq data of the primary tumor of ccRCC with complete information on age, gender, AJCC TNM staging, and survival to eliminate the interference of other factors. In addition, we excluded patients who had a survival time of less than 30 days, whose samples were collected after neoadjuvant treatment, or whose pathological type was other than ccRCC. Using these criteria, we obtained 246 ccRCC tissue samples from TCGA as a training set. All bulk RNA-seq differential analyses in this study utilized original count data from LUAD, while other analyses employed log(TPM + 1) formatted data. In parallel, after processing and annotating the E-MTAB-1980 dataset using the limma package, we obtained 101 ccRCC microarray samples as a validation set (see Table 2).

### 4.4. Mutation Data Acquisition and Processing

We used the TCGAbiolinks package [55] to download the latest TCGA ccRCC simple nucleotide variation (SNV) and copy number variation (CNV) data from the GDC. After intersecting with transcriptomic samples, we ultimately included 152 SNV samples and 273 CNV samples for gene mutation analysis and tumor mutational burden (TMB) assessment. SNV data were analyzed using the maftools package [58] to calculate TMB for nine mutation types: Frame_Shift_Del, Frame_Shift_Ins, In_Frame_Del, In_Frame_Ins, Missense_Mutation, Nonsense_Mutation, Nonstop_Mutation, Splice_Site, and Translation_Start_Site. CNV data were preliminarily processed using the GISTIC2.0 tool in GenePattern (https://cloud.genepattern.org/gp/pages/index.jsf) (accessed on 22 April 2022) [59] and subsequently analyzed using the maftools package as well.

### 4.5. Methylation Data Acquisition and Processing

We employed the TCGAbiolinks package to download the latest TCGA ccRCC Illumina Human Methylation 450 microarray data from the GDC. After overlapping with transcriptomic samples, we ultimately included methylation data from 141 samples for analysis. The impute package was used to fill in missing values, and the ChAMP package [60] was utilized for quality control filtering of methylation data and identification of differentially methylated genes.

### 4.6. Single-Cell Data Processing

We used the original UMI counts data for single-cell analysis, with preprocessing, quality control, normalization, and dimensionality reduction clustering performed using Seurat v4.0 [61]. Quality control standards included expression of each gene in at least three cells, expression of at least 500 genes in each cell, number of genes and counts in each sample based on median ± 3 * median absolute deviation (MAD) standard screening, the proportion of mitochondrial genes with 20% threshold, the proportion of hemoglobin genes with 1% threshold, and duplicate cells filtered using the DoubletFinder package [62]. Subsequent data normalization, identification of highly variable genes, and dimensionality reduction clustering were performed based on Seurat’s default parameters and standard workflow. We used the harmony package to integrate data from different samples. Cell cluster naming was performed by collecting marker genes from literature and manual annotation. The FindAllMarkers function was used to calculate differentially expressed genes between cell subpopulations (immune cell-related differentially expressed genes, ICRDEGs) based on the Wilcoxon test, with a selection criterion of adj. *p* value < 0.05 and |log2FC| > 0.5.

To identify ferroptosis-related immune-cell subpopulations, we used the AUCell package [63] to enrich activity scores of ferroptosis pathways within immune cell subpopulations. We then extracted the immune cell population with the highest scores for further subgrouping.

### 4.7. Pseudotime Analysis

We utilized the well-established monocle2 package [64] to perform pseudotime analysis on cell subpopulations that showed the highest association with ferroptosis. The single-cell data were processed by creating monocle objects, performing normalization, and filtering out low-quality cells. By selecting highly variable genes, we reduced the dimensionality of the data using the DDRTree approach. We then performed pseudotime analysis of different cell types and essential genes after ordering the cells.

### 4.8. Identification of Immune-Related Ferroptosis Genes

To discover immune-related ferroptosis genes (IRFGs), we used the estimate package [64,65] to assess immune-cell infiltration scores within different TCGA ccRCC samples. Based on the median value, we divided the ccRCC data into high and low immune infiltration groups. Subsequently, we utilized the DESeq2 package [66] to calculate differentially expressed genes between high and low immune infiltration groups (immune subtype-related differentially expressed genes, ISRDEGs), with a selection criterion of |logFC| > 0.5 and adj. *p* value < 0.05. Finally, we obtained IRFGs by intersecting ISRDEGs, ICRDEGs, and FGs.

### 4.9. Mutations and Methylation of IRFGs

To further investigate the SNV and CNV mutation profiles of IRFGs, we utilized the maftools package to generate SNV and CNV mutation maps for IRFGs in different samples. By comparing the methylation levels of IRFGs between high and low immune infiltration groups using the ChAMP package, we generated volcano plots and heatmaps illustrating the methylation differences of IRFGs between the two groups.

### 4.10. Constructing of the IRFGRs and Exploring Their Role in Prognosis

To elucidate the relationship between IRFGs and the overall survival of ccRCC patients, we used univariate and multivariate Cox regression analyses to identify independent risk factors among IRFGs and constructed an IRFGs risk score model. The calculation of the immune-related ferroptosis genes risk score (IRFGRs) is as follows:IRFGRs=β1×expG1+β2×expG2+…+βn×expGn

Subsequently, we normalized the data to provide a consistent standard. We utilized the survivalROC package [67] to evaluate the predictive performance of IRFGRs on the survival of ccRCC patients in both the TCGA training set and the ArrayExpress validation set, and generated survival curves accordingly.

To further investigate whether IRFGRs is an independent risk factor for ccRCC, we again used univariate and multivariate Cox regression analyses to comprehensively evaluate the impact of IRFGRs and other clinicopathologic characteristics on the overall survival of ccRCC patients. We constructed nomograms using the rms package, calculated the area under the curve (AUC) for the effect of different factors on survival using the survivalROC package, and validated the stability of the nomograms with calibration curves. Additionally, we used the ggDCA package to evaluate the clinical decision curves of each independent risk factor, exploring the benefit to patients in different scenarios.

### 4.11. Weighted Gene Correlation Network Analysis

Given that IRFGs are independent risk factors for overall survival in ccRCC, we employed weighted gene correlation network analysis (WGCNA) [68] to identify gene modules and genes associated with IRFGs. WGCNA is a systems biology method used to describe gene association patterns across different samples, identify highly co-varying gene sets, and determine potential genes or therapeutic targets based on the intramodular connectivity of gene sets and their associations with clinical features. We used the expression matrix of differentially expressed genes between high and low immune infiltration groups as input files and calculated the optimal soft-thresholding power with the pickSoftThreshold function. With this power, we constructed a scale-free network, calculated the topological matrix, and performed hierarchical clustering. We set the minimum module gene count to 30 to construct gene modules and merged similar modules by setting the minimum module distance to 0.2. We established correlations between each module and clinical features by correlation analysis, selected the module with the highest correlation with IRFGs as the core module, and selected the genes with the highest module association as IRFGs-related genes.

### 4.12. Relationship between Immune Cell Enrichment Scores and IRFGs

CIBERSORT [69] is an immune-cell infiltration estimation tool that can be used to assess the abundance of constituent cell types within mixed cell populations using gene expression data. We used the CIBERSORT R script to evaluate ccRCC data in TCGA and analyzed the enrichment scores of 22 immune cell types in different samples. Furthermore, we utilized the corrplot package to generate a heatmap showing the correlations between the 22 immune cell types and IRFGs, as well as their constituent genes.

### 4.13. Enrichment Analysis

Gene ontology (GO) [70] defines concepts used to describe gene functions and the relationships between these concepts, and divides gene functions into three aspects: biological process (BP), molecular function (MF), and cellular component (CC). The Kyoto Encyclopedia of Genes and Genomes (KEGG) [71] is a collection of pathway maps representing molecular interactions and reaction networks covering a wide range of biochemical processes.

The clusterProfiler package [72] is a comprehensive R package that allows GO and KEGG analysis of the given data, with enrichment criteria set at *p* value < 0.05 and *q* value < 0.05. The *p* value correction method used is Benjamini-Hochberg (BH).

### 4.14. Statistical Analysis

In this study, all data calculations and statistical analyses were performed using R software (https://www.r-project.org/, version 4.1.2) (accessed on 30 December 2021). To compare two groups of continuous variables, the Mann-Whitney U test (also known as the Wilcoxon rank-sum test) was used to analyze the differences between non-normally distributed variables. Unless otherwise stated, correlation analyses were performed using Spearman’s correlation analysis in the cor function of the R base package. Univariate and multivariate Cox regression analyses were mainly performed using the survival package, AUC calculations were based on the survivalROC package, forest plots were generated using the forestplot package, and nomogram plots were generated using the rms package. All statistical *p* values were two-sided, and for single-cell and bulk RNA-seq differential gene selection, a *p* value < 0.05 or an adj. *p* value < 0.05 was considered statistically significant. For other statistical tests, the pvalue or adj. *p* value criteria were as described in the text.

## 5. Conclusions

In conclusion, our single-cell multiomics analysis revealed that tumor-infiltrating immune cells in the TME regulate ferroptosis signaling pathways. Furthermore, a scoring scheme called the IRFGRs was constructed, tested, and found to be positively correlated with immune-cell infiltration and prognosis in patients with ccRCC. The use of the IRFGRs scoring scheme in clinical practice may provide insight into the underlying mechanisms of immune ferroptosis and subsequent TME infiltration in ccRCC patients, and may help predict prognosis in this disease.

## Figures and Tables

**Figure 1 ijms-24-09092-f001:**
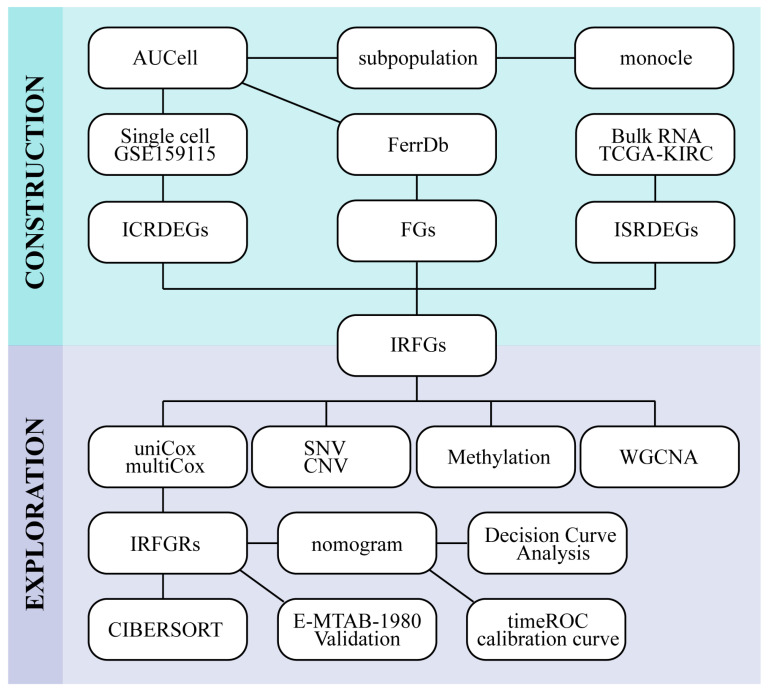
Overall analysis flow chart.

**Figure 2 ijms-24-09092-f002:**
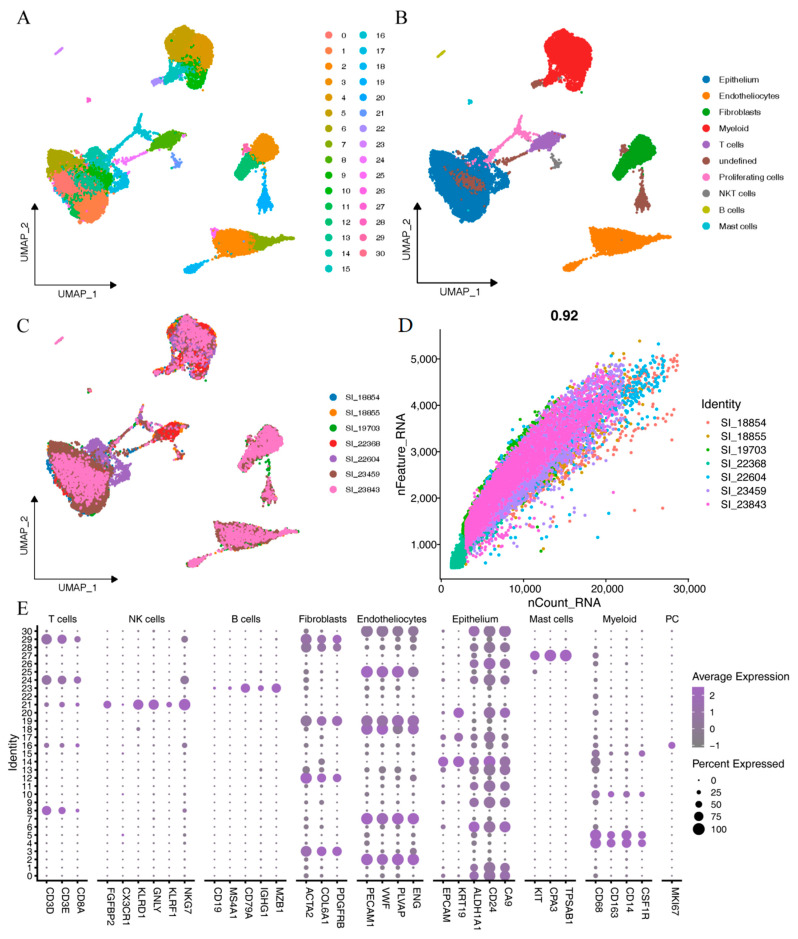
Dimensionality reduction clustering of single-cell data (GSE159115): (**A**) when resolution = 1.6, the GSE159115 dataset was divided into 31 cell populations; (**B**) in the 31 cell populations, 9 cell types were identified based on cell type-specific markers, and cells without specific markers were designated undefined; (**C**) distribution of various types of cells in 7 patients; (**D**) correlation plot of nCount and nFeature in all cells; (**E**) expression of specific markers of different cell types in 31 cell clusters.

**Figure 3 ijms-24-09092-f003:**
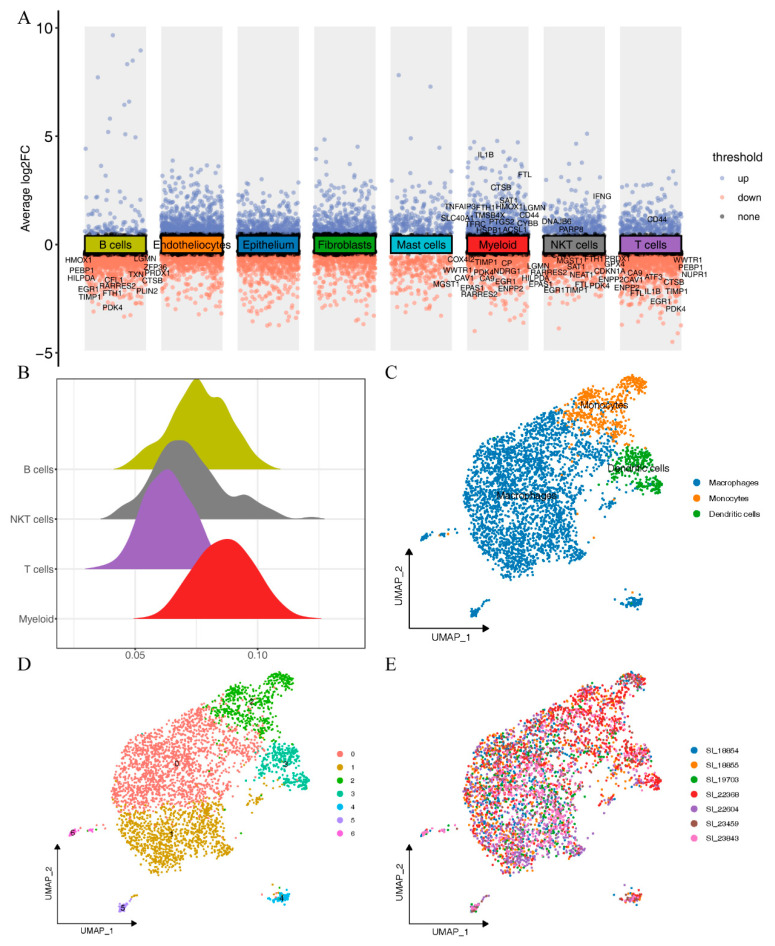
Activity of the ferroptosis pathway in different immune cells and subpopulations of myeloids: (**A**) various types of cells express different genes, and immune cells express different ferroptosis genes; (**B**) activation of 4 immune cell types along the ferroptosis pathway; (**C**) classification of seven myeloid populations based on cell type-specific markers; (**D**) when resolution = 0.2, myeloid are divided into 7 cell populations; (**E**) distribution of myeloid in 7 patients.

**Figure 4 ijms-24-09092-f004:**
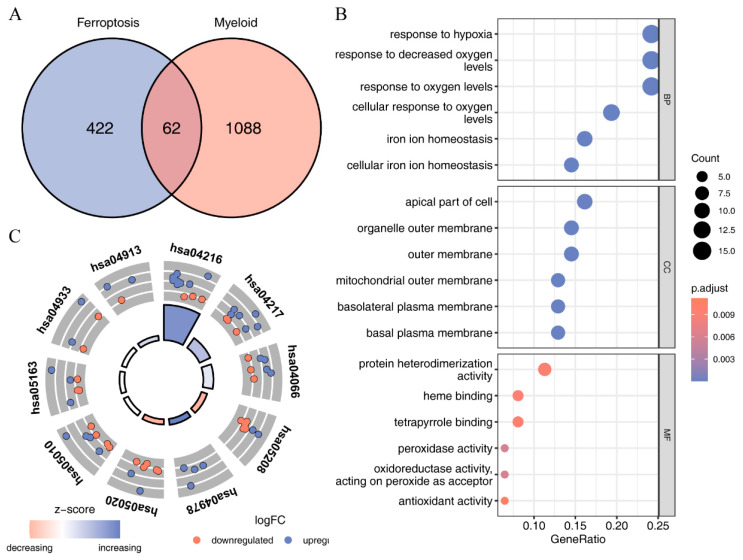
GO and KEGG enrichment analyses of 62 ferroptosis genes related to myeloids: (**A**) intersection of myeloid genes and ferroptosis genes; (**B**) the enrichment of 62 myeloid-related ferroptosis genes in the biological process (BP), cellular component (CC), and molecular function (MF) categories was analyzed using GO enrichment analysis; (**C**) pathways enriched in the 62 myeloid-associated ferroptosis genes in KEGG pathway enrichment analysis.

**Figure 5 ijms-24-09092-f005:**
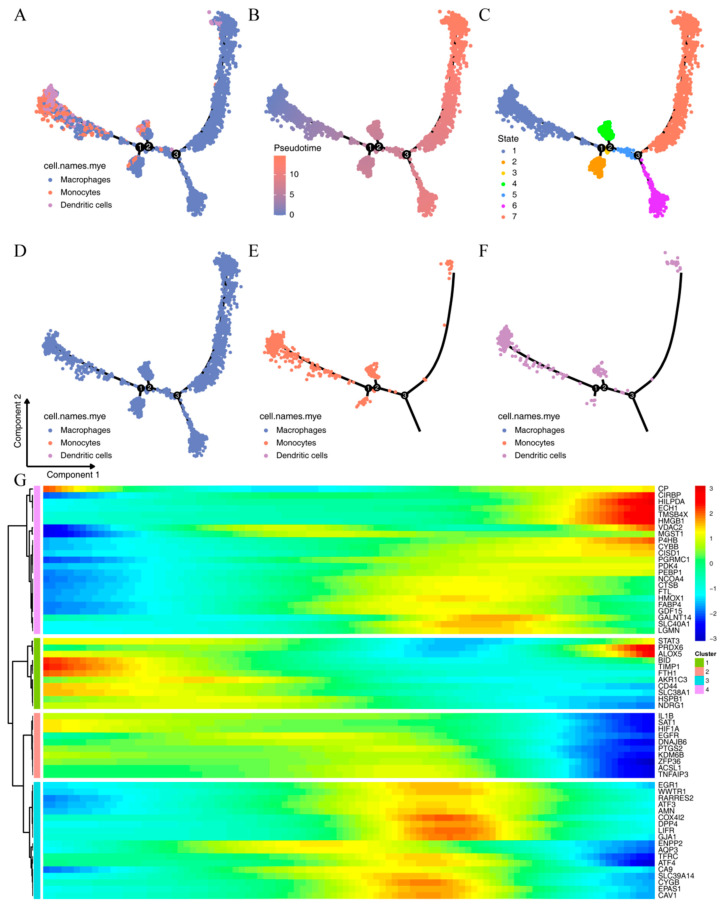
Pseudotime analysis (1, 2, and 3 in the figure represent different decision points for three possible cell fates): (**A**) distribution of three types of myeloid subsets; (**B**) pseudotime distribution trends of myeloid subsets; (**C**) different differentiation states of myeloid subsets; (**D**) distribution of macrophage subsets; (**E**) distribution of monocyte subsets; (**F**) distribution of dendritic cell subsets; (**G**) heatmap of the expression of 62 myeloid-associated ferroptosis genes over pseudotime.

**Figure 6 ijms-24-09092-f006:**
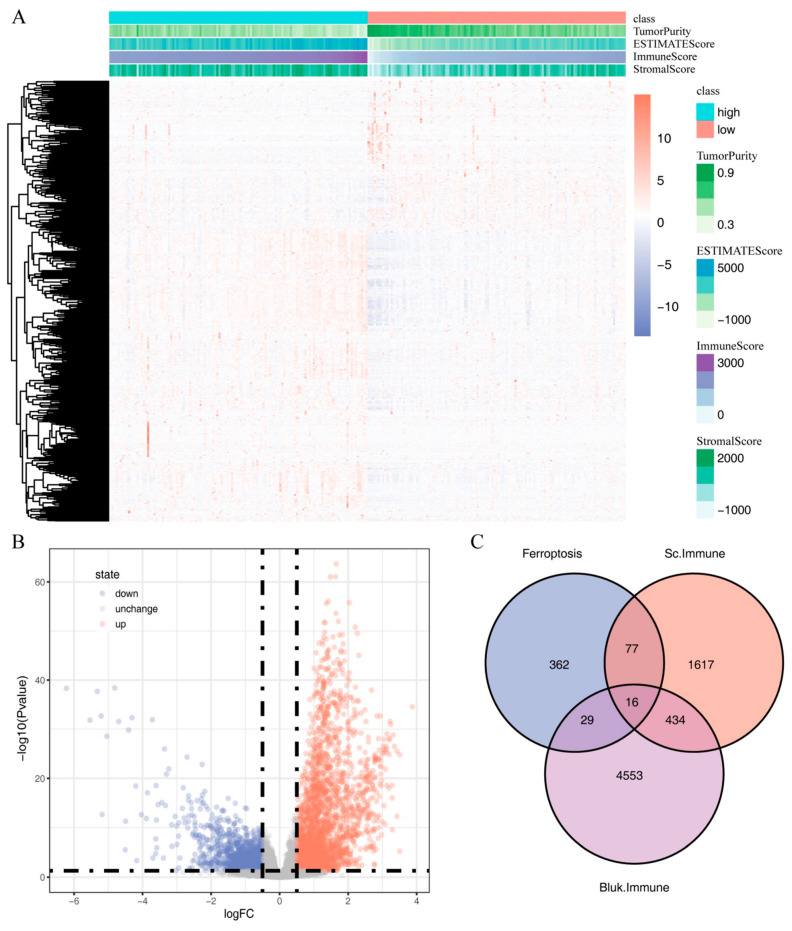
Genes and immune-related ferroptosis genes (IRFGs) that differ between samples with high and low immune-cell infiltration: (**A**) differences in gene expression between samples with high and low immune-cell infiltration; (**B**) gene expression volcano plot of samples with high and low immune-cell infiltration; (**C**) in single-cell data, IRFGs were derived from differentially expressed genes in samples with low and high immune-cell infiltration levels, as well as from genes that are associated with ferroptosis.

**Figure 7 ijms-24-09092-f007:**
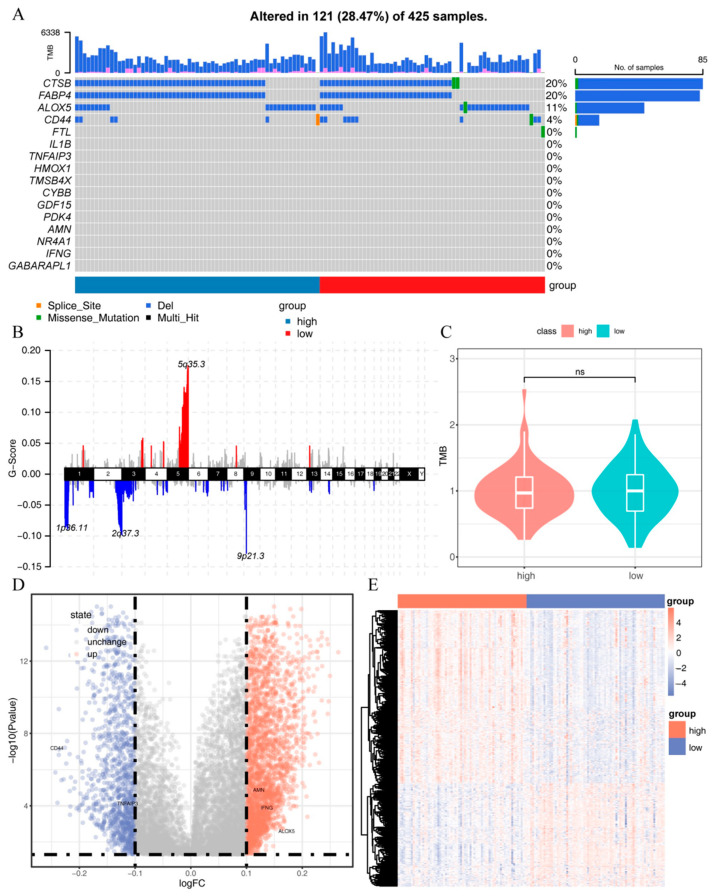
Mutation and methylation signatures of IRFGs: (**A**) SNV and CNV profiles of 16 IRFGs; (**B**) overall chromosomal copy number variation in ccRCC samples; (**C**) TMB differences between samples with high and low immune-cell infiltration; (**D**) methylation differences between samples with high and low immune-cell infiltration in a volcano plot; (**E**) methylation differences between samples with high and low immune-cell infiltration.

**Figure 8 ijms-24-09092-f008:**
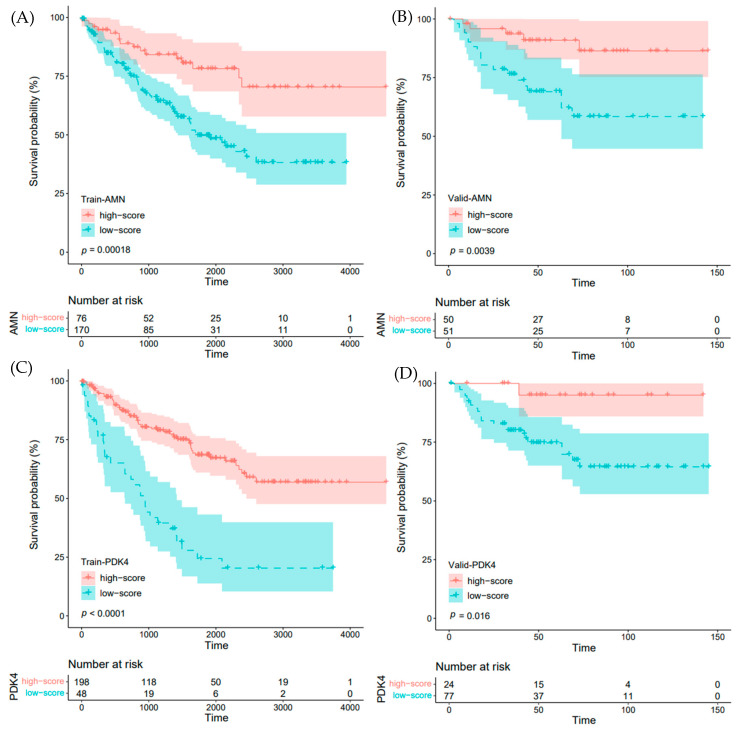
Impact of “AMN gene only” and “PDK4 gene only” on ccRCC prognosis: (**A**) impact of AMN gene on ccRCC prognosis in the TCGA dataset; (**B**) impact of AMN gene on ccRCC prognosis in the validation dataset; (**C**) impact of PDK4 gene on ccRCC prognosis in the TCGA dataset; (**D**) impact of PDK4 gene on ccRCC prognosis in the validation dataset.

**Figure 9 ijms-24-09092-f009:**
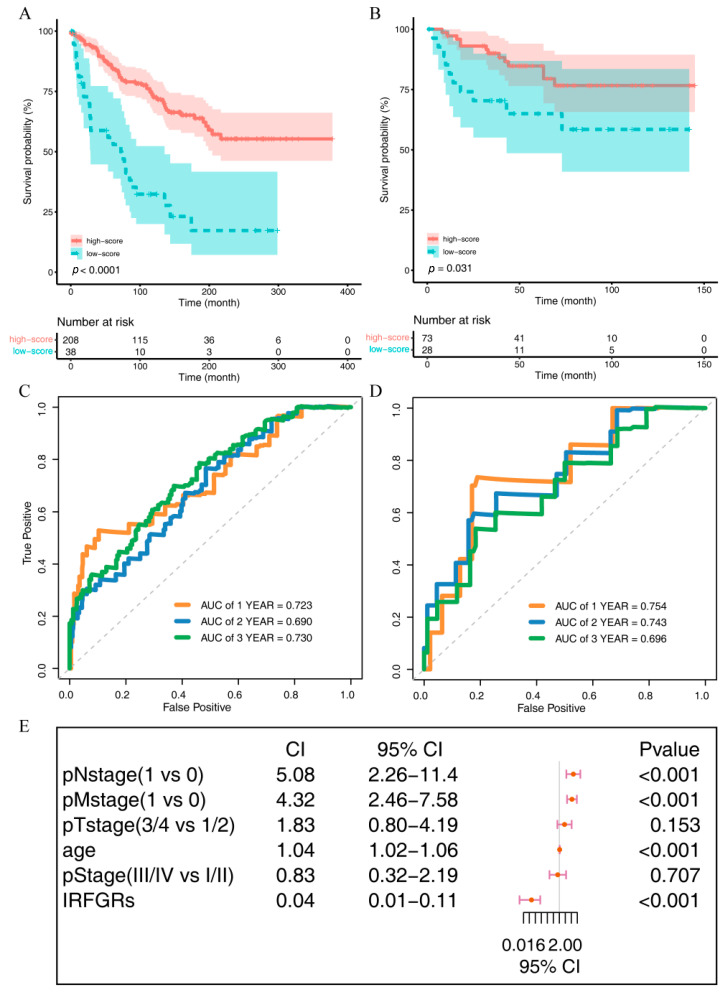
Effects of the immune-related ferroptosis genes risk score (IRFGRs) on overall survival in ccRCC: (**A**) IRFGRs survival curves in the training set; (**B**) IRFGRs survival curves in the validation set; (**C**) efficacy of the IRFGRs in predicting overall survival in the training set; (**D**) efficacy of the IRFGRs in predicting overall survival in the validation set; (**E**) results of multivariate Cox analysis of the IRFGRs combined with multiple clinicopathological indicators.

**Figure 10 ijms-24-09092-f010:**
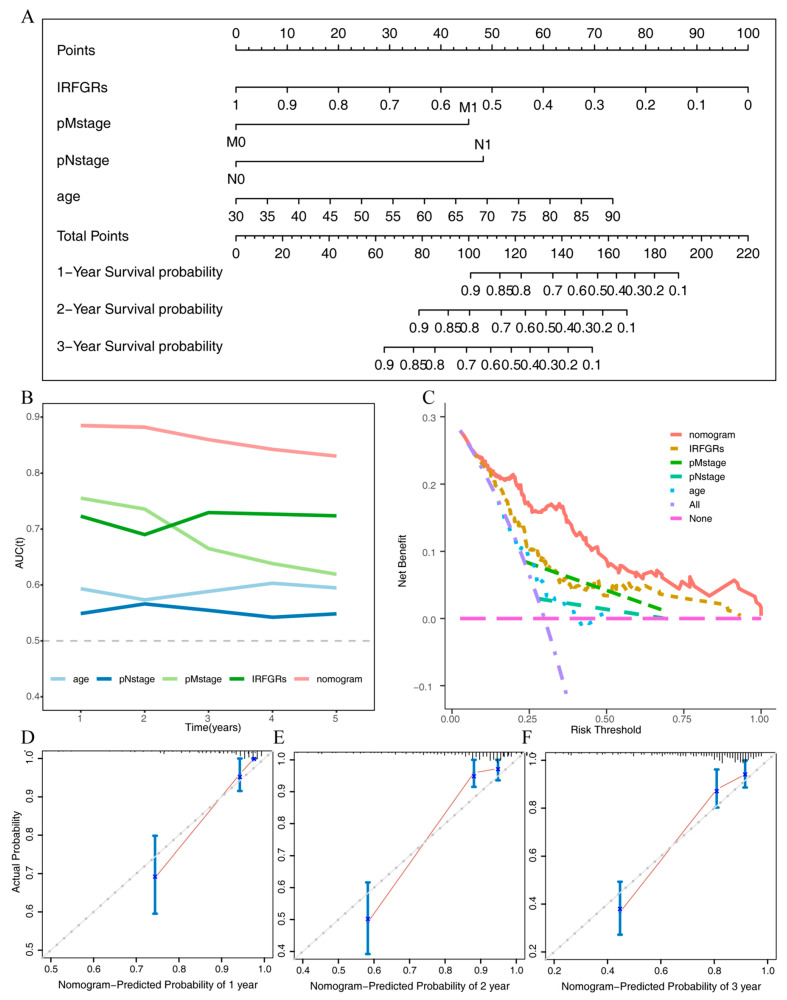
Construction of the nomogram and assessment of its efficiency: (**A**) the nomogram constructed from the IRFGRs combined with multiple clinicopathological indicators; (**B**) AUC values of independent risk factors and the nomogram for the overall survival of ccRCC in 1–5 years; (**C**) clinical decision curves of independent risk factors and the nomogram; (**D**) training set consistency curve for overall survival over one year; (**E**) training set consistency curve for overall survival over two years; (**F**) training set consistency curve for overall survival over three years.

**Figure 11 ijms-24-09092-f011:**
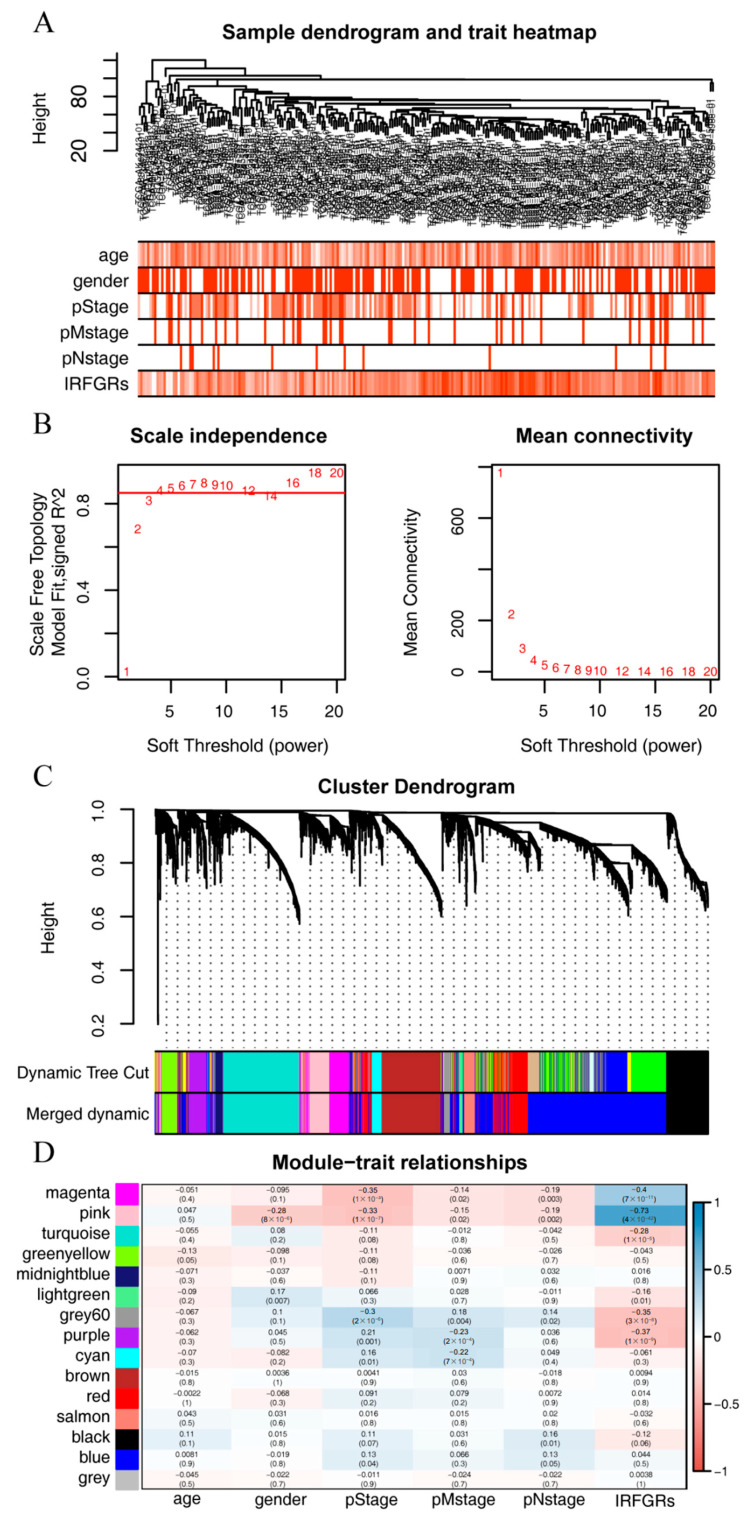
Weighted gene co-expression network analysis (WGCNA): (**A**) Elimination of outlier samples by cutting the height; (**B**) Determination of the optimal soft threshold power; (**C**) Formation and merging of modules; (**D**) Association of modular genes with the IRFGRs (positive correlations are indicated by blue, while negative correlations are indicated by red).

**Figure 12 ijms-24-09092-f012:**
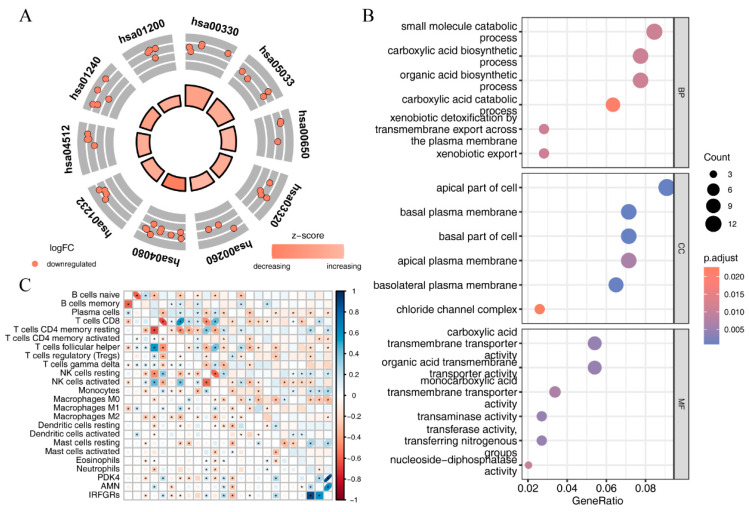
Correlation analysis between immune cell enrichment scores and the IRFGRs using GO and KEGG analyses of the pink module: (**A**) KEGG pathway enrichment analysis of pink module genes; (**B**) GO enrichment analysis of pink module genes reveals enrichment in biological processes (BP), cellular components (CC), and molecular functions (MF); (**C**) correlation analysis between immune cell enrichment scores and the IRFGRs in ccRCC samples from TCGA (darker blue means higher correlation, while darker red means lower correlation ).

**Table 1 ijms-24-09092-t001:** Patients’ baseline characteristics.

Patient	Age	Gender	pTstage	pNstage	pMstage	pStage	Treatment
1	71	Male	T2	N0	M0	Stage II	Radical nephrectomy
2	70	Male	T1	N0	M0	Stage I	Partial nephrectomy
3	69	Male	T1	N0	M0	Stage I	Radical nephrectomy
4	76	Male	T2	N0	M0	Stage II	Partial nephrectomy
5	74	Male	T2	N0	M0	Stage II	Radical nephrectomy
6	66	Male	T1	N0	M0	Stage I	Partial nephrectomy
7	71	Male	T2	N0	M0	Stage II	Radical nephrectomy

**Table 2 ijms-24-09092-t002:** Baseline characteristics.

Factors	Level	E-MTAB-1980	TCGA
		101	246
age (mean (SD))		63.48 (11.50)	61.54 (11.91)
gender (%)	female	24 (23.8)	100 (40.7)
	male	77 (76.2)	146 (59.3)
race (%)	asian	-	4 (1.6)
	black or african american	-	17 (6.9)
	unkown	-	4 (1.6)
	white	-	221 (89.8)
tumor.history (%)	no	-	212 (86.2)
	yes	-	34 (13.8)
pStage (%)	Stage I	66 (65.3)	102 (41.5)
	Stage II	10 (9.9)	34 (13.8)
	Stage III	13 (12.9)	71 (28.9)
	Stage IV	12 (11.9)	39 (15.9)
pTstage (%)	T1	68 (67.3)	106 (43.1)
	T2	11 (10.9)	42 (17.1)
	T3	21 (20.8)	93 (37.8)
	T4	1 (1.0)	5 (2.0)
pNstage (%)	N0	94 (93.1)	233 (94.7)
	N1	3 (3.0)	13 (5.3)
	N2	4 (4.0)	0 (0.0)
pMstage (%)	M0	89 (88.1)	207 (84.1)
	M1	12 (11.9)	39 (15.9)

## Data Availability

The novel findings reported in this study can be found within the article and its Appendix A. For additional information or inquiries, please contact the corresponding author.

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
