# Peer review of "Single-Cell RNA-Seq Analysis Reveals Ferroptosis in the Tumor Microenvironment of Clear Cell Renal Cell Carcinoma"

_ijms, 2023, doi:10.3390/ijms24109092_

Round 1

Reviewer 1 Report

The overall to assay the ferroptosis genes in single cell sequencing for tumor microenvironment in ccRCC is sound. But there is a big problem in picking up the key immune related ferroptosis genes (IRFGs) for ccRCC prognosis. The abstract in line 19 and line 206, it looks like only using AMN and PDK4 to calculate IRFGs score but not using all 16- IRFGs in Figure 7A. If this is true, please draw the “AMN gene only” and “PDK4 only” on ccRCC prognosis. If using all 16 genes as the final score as IRFGs score, it’s no reason to emphasis the AMN and PDK4 as the IRFGs score (also cannot find the clear reason to choose of these two genes). It also will make the future followers very hard to repeat the data as Figure 8 for calculate the 2? or 16? genes for ccRCC prognosis (so unclear and hard to know the contribution of each genes for the ccRCC).

Reviewer 2 Report

I would like to applaud the authors for their work exploring the role of ferroptosis in the immune TME of ccRCC samples.  This is a budding area of RCC research and could provide novel areas of prognostic/predictive biomarker development and therapeutic interventions.  I do feel this work contributes to the existing literature and the methods are well outlined with easy to read images.  Suggested edits are as follows:

INTRODUCTION

-Line 35: would remove conventional chemotherapy as this is not a part of standard of care for clear cell RCC; would add metastasis directed therapy (radiation, surgery) as potential therapeutic modality for ccRCC

-Line 36: unclear of the relationship between side effects and tumor heterogeneity as this is more due to patient heterogeneity, would rephrase to avoid confusion

-Line 48-49: Provide references and consider expanding on how "ferroptosis has been shown to be beneficial in treatment of ccRCC in several studies."

-Line 68: Clarify "immunological functions."

-Line 69: I agree that they help improve understanding of TME ferroptosis but I think the true benefit of these findings are that they may improve upon existing prognostic biomarkers in RCC.  

RESULTS

-Recognizing the significant disease heterogeneity, do we know any clinical details regarding the patients and their disease course for the 7 samples of ccRCC used for the study? I feel this is crucial information to include.

-Figure 4B shows highest enrichment of "myeloid related ferroptosis genes" is in hypoxia and oxygen level response.  Could this simply be explained by the known common mechanism in many ccRCC tumors of VHL loss leading to increased HIF1a activity as opposed to being associated with a novel pathway such as ferroptosis?  How specific are these pathways to ferroptosis and does limiting the results to iron ion specific pathways still demonstrate substantial ferroptosis activity specific to the myeloid cell population?

-Does the ArrayExpress validation set incorporate samples from contemporary immunotherapy era?  If not, any plans to assess the score performance in a more contemporary cohort?

DISCUSSION

-Line 293: would also make mention of CTLA4 inhibitors as approved therapy

-Line 316: Would illustrate here that your findings supported this correlation of IRFGs and M2 macrophage infiltration

-Line 336-337: Would clarify role of the genes in immunotherapy, benefit or detriment?

-Would mention a potential future direction could be to assess the predictive value of the model in terms of predicting response to immune checkpoint inhibitors as this remains an unmet need in the field (PD-1 has been shown to be poor biomarker in RCC).  Could add real clinical value to the IRFGRs model.  Would want to then know how feasible this model is for clinical use.

Minor edits only, reads clearly for the most part.

Round 2

Reviewer 1 Report

The new adding Figure 8 is much clear to support the following IRFG score model. It can be acceptable for this version of article.